# Alterations in Fecal Short-Chain Fatty Acids after Bariatric Surgery: Relationship with Dietary Intake and Weight Loss

**DOI:** 10.3390/nu14204243

**Published:** 2022-10-12

**Authors:** Jennifer L. Meijer, Meredith N. Roderka, Elsa L. Chinburg, Timothy J. Renier, Auden C. McClure, Richard I. Rothstein, Elizabeth L. Barry, Sarah Billmeier, Diane Gilbert-Diamond

**Affiliations:** 1Weight and Wellness Center, Dartmouth-Health, Lebanon, NH 03756, USA; 2Department of Medicine, Geisel School of Medicine, Dartmouth College, Hanover, NH 03755, USA; 3Department of Pediatrics, Geisel School of Medicine, Dartmouth College, Hanover, NH 03755, USA; 4Department of Epidemiology, Geisel School of Medicine, Dartmouth College, Hanover, NH 03755, USA; 5Department of Surgery, Geisel School of Medicine, Dartmouth College, Hanover, NH 03755, USA

**Keywords:** obesity, prebiotics, short-chain fatty acids, fiber, starch, protein, added sugar, bariatric surgery, weight loss

## Abstract

Bariatric surgery is associated with weight loss attributed to reduced caloric intake, mechanical changes, and alterations in gut hormones. However, some studies have suggested a heightened incidence of colorectal cancer (CRC) has been associated with bariatric surgery, emphasizing the importance of identifying mechanisms of risk. The objective of this study was to determine if bariatric surgery is associated with decreases in fecal short-chain fatty acids (SCFA), a group of bacterial metabolites of fiber. Fecal samples (*n* = 22) were collected pre- (~6 weeks) and post-bariatric surgery (~4 months) in patients undergoing Roux-en-Y gastric bypass and sleeve gastrectomy. SCFA levels were quantified using liquid chromatography/mass spectrometry. Dietary intake was quantified using 24-h dietary recalls. Using an aggregate variable, straight SCFAs significantly decreased by 27% from pre- to post-surgery, specifically acetate, propionate, butyrate, and valerate. Pre-surgery weight was inversely associated with butyrate, with no association remaining post-surgery. Multiple food groups were positively (sugars, milk, and red and orange vegetables) and inversely (animal protein) associated with SCFA levels. Our results suggest a potential mechanism linking dietary intake and SCFA levels with CRC risk post-bariatric surgery with implications for interventions to increase SCFA levels.

## 1. Introduction

Over the past three decades, the rate of adult obesity has rapidly increased in the United States, such that in New Hampshire, 9.9% of adults had obesity in 1990 compared to 29.9% in 2020 [1]. In response, medical treatments have advanced, with weight loss medications yielding 5–40% excess weight loss [2,3,4,5]. Per the National Institute of Health (NIH) guidelines, bariatric surgery is offered to patients with clinically severe obesity: a BMI ≥ 40, or a BMI ≥ 35 with obesity-related comorbidities, such as type 2 diabetes or hypertension. Within these individuals, bariatric surgery offers a larger and better-sustained weight loss than other treatments, demonstrating ~50–70% excess weight loss [6]. Bariatric surgery is associated with reduced long-term mortality [7] and improved obesity-related comorbidities including type 2 diabetes [8], hypertension [9], and sleep apnea [10]. The two most performed procedures are the laparoscopic Roux-en-Y gastric bypass (RYGB) and laparoscopic sleeve gastrectomy (LSG). Reduced caloric intake [11], mechanical changes [12], and alterations in several gut hormones (e.g., ghrelin, peptide YY and GLP-1) [13] are considered important biological drivers of weight loss. Despite the successful weight loss associated with surgery, a retrospective Swedish cohort study (*n* = 77,111) observed that bariatric surgery, performed using RYGB or adjustable gastric banding, was associated with an increased risk of colorectal cancer (CRC) [14]. Following this initial association, large epidemiology studies have yielded both positive [15,16] and null associations [17,18] between bariatric surgery and CRC, all dependent on the location of the study, number of years follow-up, and type of surgery. It is important to consider how dietary, mechanical, and hormonal alterations post-surgery may increase risk of CRC.

The primary objective of this study is to determine if bariatric surgery is associated with alterations in short-chain fatty acids (SCFA), as diminished SCFAs after bariatric surgery might suggest a mechanism linking dietary intake with CRC [19]. SCFA are a major end-product of bacterial fermentation of fiber in the human colon and have a wide range of impacts on the host physiology. For instance, butyrate has been investigated in many model systems and has been shown to promote apoptosis of cancerous cells [20]. Of interest, Farup and Valeur has reported changes in SCFA post bariatric surgery [21], demonstrating decreases in straight SCFA (e.g., acetate, propionate, and butyrate) and increases in branched SCFA (e.g., isobutyrate, isovalerate, and isocaproate). The secondary objective of this study is to determine if surgery type (RYGB vs. LSG), dietary intake, age, and weight loss are associated with changes in SCFA levels. We hypothesize that increased intake of fiber and food groups containing fiber will be positively associated with SCFA levels.

## 2. Materials and Methods

### 2.1. Cohort

Participants were recruited through Dartmouth-Hitchcock Medical Center’s bariatric surgery program in Lebanon, NH. The surgical procedure selection (RYGB or LSG) was a routine clinical joint decision by the patient and surgeon, considering each patient’s medical and surgical history. The entire surgical cohort consisted of 49 participants (RYGB *n* = 31, LSG *n* = 18). This analysis consisted of a subset of participants who provided fecal samples pre- and post-surgery (RYGB *n* = 14, LSG *n* = 8). Informed consent was obtained prior to data and specimen collection. The Dartmouth Committee for the Protection of Human Subjects approved the study.

### 2.2. Clinical Standard of Care

All participants received standard clinical care for bariatric surgery and followed the typical visit schedule with preoperative visits including consultation with surgeons, nurse practitioners and dietitians, as well as two preoperative group education classes. Immediately prior to surgery, patients completed a 2-week 800–1000 kcal low-carbohydrate diet with the objective of shrinking the liver prior to surgery. Patients were hospitalized for 1–2 days post-surgery. Postoperatively, patients were instructed to follow a standard dietary progression, with a gradual return to regular solid food by 6 weeks post-surgery. Patients followed a standard of care post-surgery visit schedule, with visits planned at 3–4 weeks, 4, 12 months. Clinical staff measured participant weight and height at each pre- and post-surgical visit. Weight and height were measured by clinical staff prior to each visit with the surgeons, nurse practitioners and dietitians, pre- and post-surgery. All weights were extracted from medical records up to one-year post-bariatric surgery.

### 2.3. Fecal Sampling

Participant fecal samples were requested immediately following consent and at 4 months post-surgery. Participants provided stool samples using the OMNIgene GUT collection kit (OMR-200 by DNA Genotek) per manufacturer instructions. This kit was provided to participants to take home and return via mail. Post collection, fecal samples are stable in the kit’s buffer for several days at room temperature [22]. Samples were subsequently stored −80 °C the CLIA-licensed and CAP-accredited Institutional Biorepository.

### 2.4. Short-Chain Fatty Acid Analysis

Targeted metabolomics was conducted by the Michigan Regional Comprehensive Metabolomics Resource Core (MRC2). Fecal samples were pre-weighed, homogenized via a probe sonicator, and centrifuged at 16,000 rpm for 10 min. 80 µL of the supernatant was transferred to a glass autosampler vial with 15 µL of 200 mM 3-nitrophenylhydrazine (3-NPH) and 15 µL of 120 mM 1-ethyl-3-(3-dimethylaminopropyl) carbodiimide (EDC)-6% pyridine. Samples were vortexed and warmed to 40 °C. 800 µL of extraction solvent was added and vortexed. The extraction solvent contained acetonitrile with an internal standards mixture, including D4 acetic acid (500 µM), D7 butyric acid (250 µM), and D11 hexanoic acid (6.25 µM). Standard curves were created by varying the concentration of a volatile fatty acid mixture (VFAM) (0 µM, 3 µM, 10 µM, 30 µM, 100 µM, 300 µM, 1000 µM, 3000 µM), which contained 10 mM of each profiled short-chain fatty acid. 80 µL of the standard solution was transferred to a glass autosampler vial with 15 µL of 200 mM 3-NPH and 15 µL of 120 mM EDC-6% pyridine. Samples were vortexed and warmed to 40 °C. 800 µL of extraction solvent was added and vortexed. The analytic platform was liquid chromatography/mass spectrometry (Agilent 6410 QQQ) with mobile phase A water + 0.1% formic acid and mobile phase B methanol +0.1% formic acid. SCFA metabolites profiled included acetate, propionate, butyrate, valerate, isobutyrate, and isovalerate. All SCFA levels were normalized to account for the weight of fecal sample.

### 2.5. Dietary Recall

Dietary habits were assessed using 24-h dietary recalls. Pre-bariatric surgery dietary recalls were collected approximately 2 months prior to surgery. Post-bariatric surgery dietary recalls were collected approximately 4 months post-surgery. The dietary recalls were self-administered on a computer or mobile device using the National Cancer Institute ASA24-2016 tool (https://epi.grants.cancer.gov/asa24/respondent/, accessed on 15 July 2021). In the event that the participants had difficulty accessing the website, a research assistant read the questions out loud to the participant and entered their responses into the website for them. The ASA24-2016 is freely available and automatically processes the input data to summarize daily total nutrients from foods and supplements. Of the 22 participants with SCFA analyzed pre- and post-surgery, 20 reported dietary intake prior to surgery and 13 reported dietary intake post-surgery.

### 2.6. Demographics

Demographic and clinical data was collected as part of the Metabolic and Bariatric Surgery Accreditation and Quality Improvement Program (MBSAQIP) [23] according to the American Society for Metabolic and Bariatric Surgery (ASMBS) [24] guidelines for program accreditation; sociodemographic information including sex, age, race, ethnicity, education, and income was assessed during clinical intake. Clinical data was abstracted from the medical records. All patients were nicotine free prior to bariatric surgery.

### 2.7. Statistics

Descriptive statistics were computed for categorical variables (Fisher’s exact test) and continuous variables (unpaired Students’ *t*-test), stratified by RYGB and LSG. LOESS regression smoothing curves were used to estimate weight loss trajectories, stratified by RYGB and LSG.

All SCFA concentrations were adjusted for weight of fecal samples. Composite variables were created as follows: straight SCFA concentration was the sum of acetate, propionate, butyrate, and valerate, branched SCFA concentration was the sum of isobutyrate and isovalerate, and total SCFA concentration was the sum of all branched and straight SCFA. Proportions of SCFA were calculated with the ratio of individual concentrations to total SCFA concentration. Individual SCFA concentrations and composite SCFA concentrations were standardized (mean 0, standard deviation 1) for statistical analyses. Comparison of pre- vs. post-bariatric surgery SCFA concentration was computed (paired *t*-test). Group differences (LSG vs. RYGB) in pre-, post-, and percent change ([post − pre]/pre × 100) SCFA concentration was computed (unpaired Students’ *t*-test). Relationship of weight closest to stool collection, age, and ASA-24 dietary intake with SCFA concentration was computed (Spearman’s rank correlation) with a significance threshold of alpha = 0.05. Hierarchical clustering of correlations yielded four distinct clusters that were described as a group. All data analysis will be performed using R version 4.0. 

## 3. Results

### 3.1. Cohort Details

Twenty-two participants provided pre- and post-surgery stool samples, with 14 undergoing RYGB and 8 undergoing LSG (Table 1). The majority of participants were white, non-Hispanic females. On average, LSG participants were younger than RYGB (RYGB: 58.6 years, LSG: 47.4 years, *p*-value = 0.018). There was no statistically significant difference in weight at time of surgery by procedure type (RYGB: 127 kg, LSG: 120 kg, *p* = 0.540). Of the 18 participants that had a weight measurement between 4 months to 1 year after surgery, there was no statistically significant difference by surgery type (RYGB: 97 kg, LSG: 99 kg, *p* = 0.889). On average, participants lost 29 ± 11 kg within the first year after surgery. Individual weight trajectories by surgery type are depicted in Figure 1a, with a LOESS smoothing function used in Figure 1b. Pre-surgery stools samples were collected on average 54 days before surgery. A trend was observed in which LSG participants mailed their post-surgery stool samples back earlier than RYGB (148 days vs. 210 days post-surgery, *p* = 0.090).

### 3.2. Short-Chain Fatty Acid Differences, Pre- and Post-Surgery

The primary objective of this analysis was to determine if levels of short-chain fatty acids differed before and after surgery. Total SCFA significantly decreased by 26% from pre- to post-surgery (*p* = 0.026) (Table 2). All profiled straight SCFA significantly decreased by 27% from pre- to post-surgery (*p* = 0.019), specifically acetate (*p* = 0.037), propionate (*p* = 0.018), butyrate (*p* = 0.023), and valerate (*p* = 0.032) (Table 2). The proportion of straight SCFA decreased (*p* = 0.026) and the proportion of branched SCFA increased (*p* = 0.026) from pre- to post-surgery (Table 2).

We analyzed differences in pre-surgery SCFA concentration by surgery type (Appendix A). Valerate was significantly higher in LSG vs. RYGB prior to surgery (*p* = 0.013), which was the only SCFA that differed by group prior to surgery. We analyzed differences in post-surgery SCFA concentration by surgery type (Appendix A). No significant differences were observed. We analyzed differences in percent change of SCFA between surgery type (Appendix A). The negative percent change of valerate in LSG was statistically significantly different than the percent change in RYGB (*p* = 0.012). Interesting, an inverse percent change for each SCFA was observed in the LSG group, which was not observed in the RYGB group.

### 3.3. Age, Weight, and Diet Correlation with SCFA Levels

The relationship between SCFA levels, pre- and post-bariatric surgery, with age, weight, and dietary intake was explored, with four distinct clusters of food type identified from hierarchical clustering (Figure 2). Significant correlations with SCFA levels were observed in Clusters 1, 2, and 4. 

Positive correlations were observed between pre-surgery SCFA levels with Cluster 1 food groups, including total sugars, milk, fruits, carbohydrates, red and orange vegetables, cheeses, and added sugars components. Total sugars (r = 0.44, *p* = 0.050), added sugars (r = 0.50, *p* = 0.024), and milk (r = 0.46, *p* = 0.040) were significantly correlated with the composite total SCFA score. Most Cluster 1 pre-surgery correlations were trending at post-surgery, potentially not reaching statistical significance due to the decrease in 24-h recall responses post-surgery (20 vs. 13 completed). Age was positively correlated with each pre-surgery individual SCFA level and the total SCFA composite score (r = 0.62, *p* = 0.002), with none of the significant correlations sustaining post-surgery.

Positive correlations were observed between post-surgery SCFA levels with Cluster 2 food groups, including starchy vegetables and whole grains. Specifically, starchy vegetables were associated with increased branched-chain SCFA (r = 0.77, *p* = 0.002) and whole grains were associated with increased acetate (r = 0.59, *p* = 0.035). No trends of Cluster 2 correlations were observed pre-surgery.

Inverse correlations were observed between pre- and post-surgery SCFA levels with Cluster 4 food groups. Meat intake was the only food group with correlations with the composite total SCFA score at both pre-surgery (r = −0.50, *p* = 0.026) and post-surgery (r = −0.65, *p* = 0.016). Weight, measured in clinic closest to stool collection date, had a trending inverse association with SCFA, only reaching significance with pre-surgery butyrate levels (r = −0.50, *p* = 0.02). Associations with all dietary food groups are reported in Appendix A.

## 4. Discussion

In this study, we have demonstrated a significant decrease in straight SCFA levels after bariatric surgery, including acetate, propionate, butyrate, and valerate. SCFA are fatty acids with fewer than six carbons, that are derived from gut microbiota fermentation of indigestible foods (e.g., polysaccharides) in the colon. Primary microbiota fermenters of polysaccharides produce acetate, lactate, and oligosaccrides [25]. Specific pathways via secondary microbiota fermenters are necessary to produce propionate, butyrate, and valerate [26]. Our results demonstrate post-bariatric surgery decreases in profiled straight SCFA via primary degraders (acetate) and secondary degraders (propionate, butyrate, and valerate) (Table 2). Each of these SCFA play a specific role in host health, with all contributing as a main source of energy for the colonocytes [27]. Branched SCFA are produced mainly through the fermentation of protein-derived branched chain amino acids. Although we did not observe increases an individual branched SCFA, we did observe an increase of the proportion of branched SCFAs (Table 2). Changes in SCFA post bariatric surgery has also been observed in one other study [21] that also found decreases in straight SCFA (e.g., acetate, propionate, and butyrate) and increases in branched SCFA (e.g., isovalerate, isobutyrate, isocaproate). Currently, our group is determining if bariatric surgery is associated with changes in the primary and secondary microbiota degraders of polysaccharides in conjunction with alterations of SCFA levels.

Dietary intake was collected from a subset of participants to determine if nutrients and food groups were associated with SCFA levels. As SCFAs are a major end-product of bacterial fermentation of fiber, we hypothesized that fiber would be positively associated with SCFA levels. These results were not observed, potentially because this cohort consumed less fiber than the recommended amount from the Nutritional Guidelines (25–38 g), with an average of 15 g prior to surgery and 13 g post-surgery (data not shown). Many individuals did not eat types of foods with high fibers, including fruits (65% of participants reported ≥ one serving), starchy vegetables (45% of participants reported ≥ one serving), whole grains (55% of participants reported ≥ one serving), and legumes (5% of participants reported ≥ one serving) (Appendix A). Furthermore, as the chemical structure of dietary fiber determines if it is accessibility to microbes [28], the dietary software used may not have been specific enough to stratify by types of fiber (e.g., resistant starches [29]) that would influence SCFA levels. Several polysaccharide rich food groups had positive correlations with SCFA prior to surgery (e.g., red and orange vegetables) and post-surgery (e.g., starchy vegetables and whole grains). However, it’s uncertain why some of the high-fiber dietary groups were associated with increased branched SCFA (e.g., starchy vegetables and total branched SCFA), as branched SCFA are derived from microbiota branched chain amino acid metabolism.

An inverse association was observed between animal protein intake and the composite total SCFA score both pre-surgery and post-surgery. Animal protein is degraded in the small intestine, while plant proteins and fibers reach the large intestine in greater quantities [30]. Degradation of protein yields the production of various bioactive molecules (e.g., ammonium and hydrogen sulfide), which could influence the composition of the gut microbiota [31]. Post-bariatric nutritional guidelines emphasize protein intake, as it is recommended that patients consume at least 60 g of dietary protein per day to attenuate loss of fat-free mass following bariatric surgery [32]. It is important to consider the colonic lumen consequences of high-protein recommendations, determining if promoting plant rather than animal protein may conserve gut microbiota structure and diversity. Furthermore, dietary intake post-bariatric surgery may be impacting the observed increased incidence of CRC [14], as high dietary fiber [33,34,35] and lower animal protein [36,37] has been demonstrated as protective against CRC incidence. The production of SCFAs may be the mediator of the association. For instance, butyrate has been investigated in many model systems as reviewed by Scharlau et al., and has been shown to reduce inflammation, inhibit tumor cell growth, induce cell differentiation, and promote apoptosis of cancerous cells [20]. In humans, observational studies have found a differential abundance of specific bacterial taxa in the stool of subjects with colorectal cancer compared with healthy controls [38,39,40], including diminished abundance of bacteria producing the short-chain fatty acid butyrate [39,41].

These results demonstrate that straight SCFA levels decrease in response to bariatric surgery, potentially due to dietary changes. In progress analyses are identifying gut microbiota species associated with bariatric surgery and SCFA production. Ongoing recruitment will increase the number of participants to determine if the mechanical differences between RYGB and LSG influence levels of SCFAs, which our small sample size may not have captured (Appendix A). Limitations of this study include the small sample size, the lack of ethnic diversity, and only have one center included in this study. Implications of the decrease in straight SCFA post-surgery needs to be elucidated. For instance, SCFAs play a vital role in communication across the gut-brain axis, stimulating the production of glucagon-like peptide-1 [42], which in turn stimulates insulin secretion and suppresses appetite [43]. However, a limitation of this study is the relevance of fecal SCFAs compared with levels of colonic SCFA or blood SCFA. Future work should consider both fecal and plasma SCFA. It is necessary to consider how to mitigate the decreases in SCFA levels post-surgery to support metabolic health. Two recent studies [44,45] have demonstrated that a short-term dietary intervention using a resistant starch supplement can alter the gut microbiome composition and increase levels of short-chain fatty acids in the stool. Future work will consider using a resistant starch dietary supplement post-surgery. Future work will prioritize complete 24-h recalls responses, using a platform with a more comprehensive nutrient database for more detailed assessment of food components. Our group will also consider the influence of race and sex, as recruitment for this study was very homogenous. This work may lead to a clinical trial testing if a dietary intervention can modify SCFA levels post-bariatric surgery, potentially leading to beneficial health effects such as decreased risk of CRC.

## Figures and Tables

**Figure 1 nutrients-14-04243-f001:**
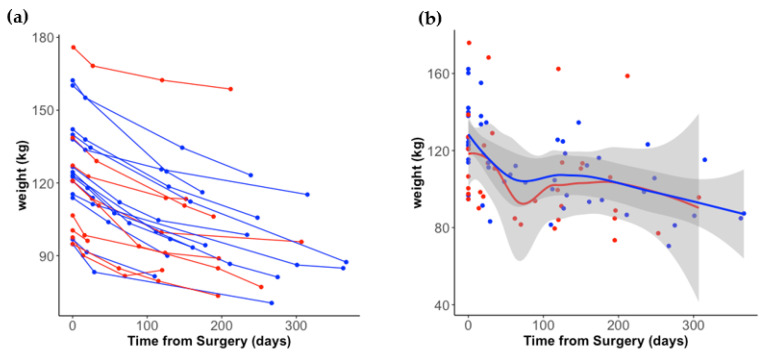
Weight loss trajectory post-surgery, stratified by surgery type. (**a**) Individual levels by surgery type are depicted by colored dots and lines: RYGB (blue) and LSG (red). (**b**) LOESS smoothing curve applied by group.

**Figure 2 nutrients-14-04243-f002:**
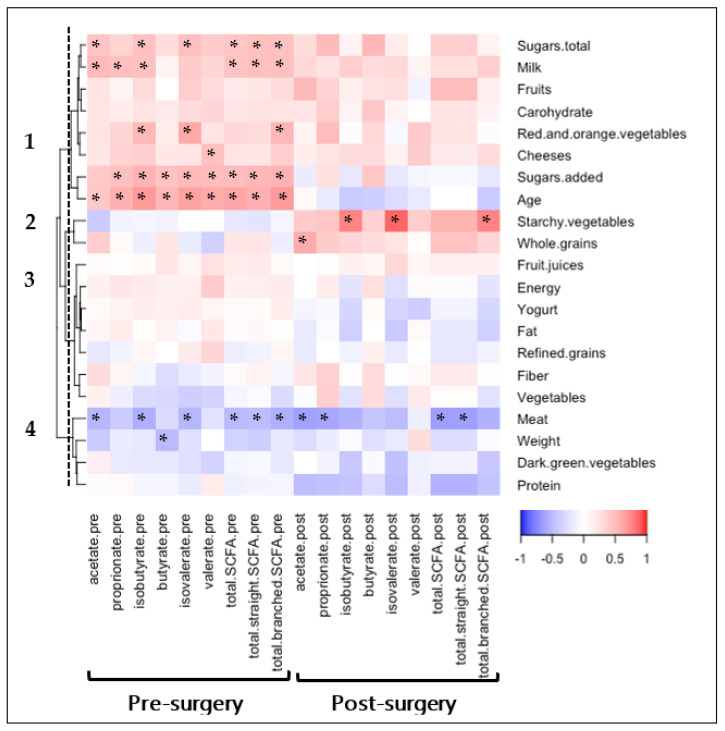
Relationship between age, weight, and diet with SCFA levels, pre- and post-bariatric surgery. Age, weight, and diet variables were collected both pre- and post-bariatric surgery. Pre-surgery variables are compared with pre-surgery SCFA. Post-surgery variables are compared with post-surgery SCFA. Spearman’s rank correlation coefficients depict positive (red) or inverse (blue) relationship with significance reported with asterisks “*” (*p* < 0.05). Variables are clustered using hierarchical clustering with cluster numbers listed.

**Table 1 nutrients-14-04243-t001:** Characteristics of study participants, stratified by surgery type. Significance denoted with unadjusted *p*-value < 0.05 (bolded).

Categorical Variables	RYGB	LSG	*p*-Value ^1^
	*n* (%)	*n* (%)	
*n*	14	8	
Sex			
male	0 (0%)	1 (13%)	0.3636
female	14 (100%)	7 (87%)	
Race			
Asian/Pacific Islander	0 (0%)	0 (0%)	1.000
African American/Black	0 (0%)	0 (0%)	
White	13 (93%)	8 (100%)	
More than one race	0 (0%)	0 (0%)	
Did not wish to report	1 (7%)	0 (0%)	
Ethnicity			
Hispanic	13 (93%)	8 (100%)	1.000
Non-Hispanic	0 (0%)	0 (0%)	
Did not wish to report	1 (7%)	0 (0%)	
**Continuous Variables**	**RYGB**	**LSG**	***p*-Value ^2^**
	**Mean ± SD (*n*)**	**Mean ± SD (*n*)**	
Age (years)	47.4 ± 10.6 years (14)	58.6 ± 9.10 years (8)	**0.018**
Pre-Surgery Body Mass Index (BMI)	48.1 ± 8.4 m^2^/kg (14)	42.5 ± 8.6 m^2^/kg (8)	0.162
Weight			
Pre-Surgery Weight	130 ± 20 kg (14)	126 ± 28 kg (7)	0.738
Bariatric Surgery Weight	127 ± 20 kg (14)	120 ± 27 kg (8)	0.540
Post-Surgery Weight (0 days–1 month)	120 ± 23 kg (9)	117 ± 29 kg (6)	0.859
Post-Surgery Weight (1 month–4 months)	106 ± 15 kg (9)	104 ± 31 kg (6)	0.865
Post-Surgery Weight (4 months–1 year)	97 ± 16 kg (11)	99 ± 31 kg (7)	0.886
Percent Change of Weight	−25 ± 6% (14)	−20 ± 9% (8)	0.193
Timing of Weight Collection (days from surgery)			
Pre-Surgery Weight	−52 ± 32 days (14)	−72 ± 44 days (7)	0.319
Post-Surgery Weight (0 days–1 month)	22 ± 5 days (9)	22 ± 7 days (6)	0.942
Post-Surgery Weight (1 month–4 months)	108 ± 25 days (9)	116 ± 13 days (6)	0.505
Post-Surgery Weight (4 months–1 year)	256 ± 71 days (11)	224 ± 71 days (7)	0.360
Percent Change of Weight	250 ± 97 days (14)	211 ± 76 days (8)	0.310
Timing of Stool Collection (days from surgery)			
Pre-Stool Collection	−47 ± 35 days (14)	−67 ± 38 days (8)	0.230
Post-Stool Collection	210 ± 51 days (14)	148 ± 83 days (8)	0.090

^1^ Represents Fisher’s exact test for categorical variables. ^2^ Represents unpaired *t*-test for continuous variables.

**Table 2 nutrients-14-04243-t002:** Short-chain fatty acid response to bariatric surgery in all participants. Significance denoted with unadjusted *p*-value < 0.05 (bolded). Represents paired *t*-test.

	Pre-Surgery	Post-Surgery	Pre- vs. Post-Surgery
	Mean ± SD	Mean ± SD	*p*-Value
Raw values (mmol/kg fecal weight)			
Total SCFA	23.1 ± 12.1	17.2 ± 6.6	**0.026**
Acetate	14.1 ± 7.9	10.9 ± 4.9	**0.037**
Propionate	3.5 ± 1.9	2.3 ± 1.3	**0.018**
Butyrate	3.1 ± 1.7	2.1 ± 1.1	**0.023**
Valerate	0.8 ± 0.5	0.5 ± 0.2	**0.032**
Isobutyrate	0.8 ± 0.5	0.7 ± 0.3	0.567
Isovalerate	0.6 ± 0.4	0.5 ± 0.2	0.127
Straight SCFA	21.5 ± 11.4	15.8 ± 6.2	**0.019**
Branched SCFA	1.5 ± 0.9	1.4 ± 0.6	0.567
Proportions (%)			
Acetate	61.0 ± 6.0	63.0 ± 9.0	0.403
Propionate	15.0 ± 3.0	13.0 ± 4.0	0.072
Butyrate	13.0 ± 3.0	12.0 ± 5.0	0.233
Valerate	4.0 ± 3.0	3.0 ± 1.0	0.446
Isobutyrate	3.0 ± 1.0	4.0 ± 1.0	**0.026**
Isovalerate	3.0 ± 1.0	3.0 ± 1.0	0.616
Straight SCFA	93.0 ± 2.0	92.0 ± 1.0	**0.026**
Branched SCFA	7.0 ± 2.0	8.0 ± 3.0	**0.026**

## Data Availability

The data presented in this study are available on request from the corresponding author. The data are not publicly available due to privacy concerns.

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
