# Peer review of "Alterations in Fecal Short-Chain Fatty Acids after Bariatric Surgery: Relationship with Dietary Intake and Weight Loss"

_nutrients, 2022, doi:10.3390/nu14204243_

Round 1
Reviewer 1 Report
This study investigated the change of fecal short-chain fatty acids pre and after bariatric surgery or weight loss surgery, which could be a risk factor for colorectal cancer. Overall, the study is well presented. Some minors are suggested.
Authors should list the limitation of the current study, such as the number of patients and one-center studies.
Abbreviations should be checked, such as SCFAs or short-chain fatty acids (alternative use in the manuscript), NPH, and EDC.
uL > μL and number and unit should be separated, such as 200mM > 200 mM.
Font of words should be consistent.
Author Response
1. Authors should list the limitation of the current study, such as the number of patients and one-center studies.
Thank you for this comment. Please see addition of limitation in lines 285-286 and 290-292.
2. Abbreviations should be checked, such as SCFAs or short-chain fatty acids (alternative use in the manuscript), NPH, and EDC.
All abbreviations were checked, and small edits were made. “Short-chain fatty acid” was used. We kept the abbreviation for short-chain fatty acid as “SCFA” or the plural “SCFAs.” In the metabolomics sample processing section, 3-NPH is 3-nitrophenylhydrazine and EDC is 1-ethyl-3-(3-dimethylaminopropyl) carbodiimide.
3. uL > μL and number and unit should be separated, such as 200mM > 200 mM.
All edits were made.
4. Font of words should be consistent
Small formatting edits were made to ensure the font of the words was consistent.
Reviewer 2 Report
The title is suggestive of the work presented
The objective of this study is relevant highlighting is to determine if bariatric surgery is associated with alterations in short-chain fatty acids (SCFA), as diminished SCFAs after bariatric surgery might suggest a mechanism linking dietary intake with CRC
The manuscript is clear, relevant to the field, and presented in a well-structured manner
The manuscript is scientifically sound and is the experimental design appropriate to test the hypothesis
The results of the manuscript are reproducible based on the details provided in the methods section.
The conclusions are consistent with the evidence
Author Response
Thank you for reviewing our manuscript. We appreciate you positive comments.